# Application of Unmanned Aerial Vehicle (UAV) Sensing for Water Status Estimation in Vineyards under Different Pruning Strategies

**DOI:** 10.3390/plants13101350

**Published:** 2024-05-13

**Authors:** Juan C. Nowack, Luz K. Atencia-Payares, Ana M. Tarquis, M. Gomez-del-Campo

**Affiliations:** 1CEIGRAM, ETSIAAB, Universidad Politécnica de Madrid (UPM), 28040 Madrid, Spain; j.nowack@hotmail.com (J.C.N.); lkatenciapayares@gmail.com (L.K.A.-P.); maria.gomezdelcampo@upm.es (M.G.-d.-C.); 2Departamento de Producción Agraria, ETSIAAB, Universidad Politécnica de Madrid (UPM), Av. Puerta de Hierro, n° 2–4, 28040 Madrid, Spain; 3Unmanned Technical Works (UTW), Leganés, 28919 Madrid, Spain; 4Grupo de Sistemas Complejos ETSIAAB, 28040 Madrid, Spain

**Keywords:** canopy development, *Vitis vinifera*, production, vegetation index, chlorophyll

## Abstract

Pruning determines the plant water status due to its effects on the leaf area and thus the irrigation management. The primary aim of this study was to assess the use of high-resolution multispectral imagery to estimate the plant water status through different bands and vegetation indexes (VIs) and to evaluate which is most suitable under different pruning management strategies. This work was carried out in 2021 and 2022 in a commercial Merlot vineyard in an arid area of central Spain. Two different pruning strategies were carried out: mechanical pruning and no pruning. The stem water potential was measured with a pressure chamber (Ψ_stem_) at two different solar times (9 h and 12 h). Multispectral information from unmanned aerial vehicles (UAVs) was obtained at the same time as the field Ψstem measurements and different vegetation indexes (VIs) were calculated. Pruning management significantly determined the Ψ_stem_, bunch and berry weight, number of bunches, and plant yield. Linear regression between the Ψ_stem_ and NDVI presented the tightest correlation at 12 h solar time (R^2^ = 0.58). The red and red-edge bands were included in a generalised multivariable linear regression and achieved higher accuracy (R^2^ = 0.74) in predicting the Ψ_stem_. Using high-resolution multispectral imagery has proven useful in predicting the vine water status independently of the pruning management strategy.

## 1. Introduction

Pruning is usually performed to control the vine’s vegetative development and generally implies a reduction in the leaf area, vigour and reserve accumulation compared to a non-pruned vine. A reduction in vegetative material can lead to more spaces in the canopy and exposed bunches [1]. Vinegrowers can adopt the no-pruning vineyard management practice to reduce operating costs and grape size [2], increasing the skin to pulp relation, an interesting feature for quality winemaking as it contributes to higher tannin and anthocyanin levels. However, it generally leads to an increment in the total leaf area [3], thus to greater transpiration levels [4] and water uptake demands, which, if not fulfilled, could lead to physiological water stress.

There is great interest in determining the plant water status in irrigated vineyards due to its relationship with the yield, fruit composition and wine quality [5,6], critical parameters for a profitable winemaking company. Water stress reduces photosynthetic activity and vegetative growth and limits berry ripening [7,8] The stem water potential (Ψ_stem_) is a consistent and sensitive indicator of the plant water status in grapevines [9], and diverse authors have published optimal thresholds of Ψ_stem_ for different phenological stages [10]. It is an integrating measurement that can provide greater precision than the soil water content concerning irrigation management. However, determining the Ψ_stem_ in commercial vineyards has a series of downsides. Only small samples can be assessed rapidly because the Ψ_stem_ is a changing parameter throughout the day. The maximum Ψ_stem_ (higher vine hydration) occurs before dawn and starts descending as the plant transpires throughout the day [11]. To compare the Ψ_stem_ values from different treatments, they must be assessed in the shortest possible time. 

Moreover, it is highly time-consuming, making it unpractical for large plantations where the intra-field variability is usually high. This evidences the necessity of developing a well-founded method for determining the grapevine water status in a cost-effective manner. 

Remote sensing is a powerful tool increasingly being used at the commercial and research levels as it can obtain vast amounts of valuable and accurate geospatial data on a large-scale dimension. Some practical applications include the discrimination of plant species or vegetation types or the detection of diseased or physiologically stressed plants [12]. Precision viticulture, a recently acquired term, can be described as the precision agriculture sector focusing on the vineyard. Its main objective is to use diverse technologies to manage vineyard spatial heterogeneity to reduce environmental impacts while increasing profitability. This spatial and temporal variability can be expressed through productivity, vine development, water status, or exogenous factors such as the soil characteristics or microclimate conditions. Some common applications of remote sensing in vineyard management include the assessment or estimation of the chlorophyll and carotenoid concentrations [13,14], grape phenolic content [15] or colour [16], canopy structure [17,18] and water status [19,20,21].

With the use of vegetation indexes (VIs), a wide range of particular characteristics, like the vegetation biomass, productivity, biochemical properties or crop water status of a photosynthetically active plant, can be assessed based on the plant spectral response. Ref. [22] conducted an extensive review of the application of remote sensing-derived vegetation indexes (VIs) in viticulture, with 113 publications evaluated since 2000. They discovered that the most commonly used platforms are currently unmanned aerial vehicles (UAVs), aircraft, and Sentinel 2 satellites. The pursued objective and the imagery’s price and resolution mainly conditioned each platform’s utilisation. While commercial satellites can be suitable for regional-scale studies due to their extensive coverage [23], their generally low and inflexible spatial and temporal resolutions make them unattractive for managing at a vineyard or site-specific scope. 

Concerning water variability management, ref. [24] used high-resolution multispectral and thermal sensors mounted on a UAV to estimate the water status in a rain-fed Tempranillo vineyard. They found that specific spectral indexes were significantly correlated to the Ψ_stem_ and stomatal conductance, both water status indicators, using 10 cm/pixel images. The same occurred with thermal indexes derived from 30 cm/pixel thermal images. They stated that thermal imagery could be helpful as a short-term water stress indicator, considering that the correlations changed throughout the season. Conversely, multispectral indexes can serve as long-term indicators as their correlations are more stable. Other related studies, like the one in [21], obtained similar results. They used 1 m/pixel multispectral images to prove that zones based on the Normalised Difference Vegetation Index (NDVI), one of the most well-known and -used VIs, values presented significant differences in the vine vegetative development, yield, and water status. Significant correlations between this index and the grapevine yield were also found by [5] using higher spatial resolution (2.6 cm/pixel) only 40 days before harvest. In general, the VIs are related to different agronomic parameters, such as the leaf area development, determined by pruning technique, which may modify vine water status and, therefore, the VI values. The leaf area is considered one of the most important agronomic parameters for evaluating the vegetative development of the plants; however, leaf area measurements are time-consuming and labour-intensive because of the inherent variability found within a vine and, on a larger scale, within a vineyard. Some methods are destructive and cannot be performed on a large scale [25]. On the other hand, the pruning weight is a parameter that encompasses the final performance of the plant. As the percentage of soil coverage, the pruning weight can differentiate between treatments, making it a good indicator in terms of the development of the vines. Other studies have observed a direct relationship between the yield through the pruning weight and the percentage of soil coverage [26,27,28]. The percentage of soil coverage appears to be an interesting tool as a proxy for the vegetative development and yield. 

Other works have focused on using spectral proximal-sensing devices instead of UAVs. For example, ref. [29] evaluated hyperspectral reflectance indexes derived from a spectrometer (350–2500 nm region) to detect grapevine water status. Their study evidenced the existence of indexes capable of significantly correlating to water status parameters (both at canopy and leaf level), such as the total leaf water content, Ψ_stem_ or equivalent water thickness. 

The above-mentioned studies and others will serve as a reference, considering that the availability of a varied set of spectral bands conditions the use of multiple VIs.

It must be emphasised that applying remote sensing to a discontinuous crop, like vineyards or fruit orchards, is technically more challenging than to a continuous herbaceous crop [30,31,32]. From an aerial perspective, the presence of the soil layer, which could be vegetated or not, does not allow for the use of widely accessible low-spatial-resolution satellite scenes (10 m/pixel for Sentinel-2). If the pixel size is excessive, the soil or intra-row vegetation reflectance deteriorates the data quality, which would not represent the canopy’s status. In general terms, the pixel size should not be coarser than the targeted object or unit, an individual vine in this study. However, some studies have proven helpful in estimating the vine water status in Mediterranean climates with nanosatellite-based imagery (3 m/pixel) by compensating for the low spatial resolution with high temporal availability [33]. Until high-resolution satellite information is achieved and freely accessible, using UAVs with multispectral sensors remains one of the vineyard’s best technical solutions for remote sensing, as pure canopy pixel information can be extracted. The pixel size may vary depending on the intended spatial coverage, but it can be down to 1.4 cm/pixel for monitoring grapevines. 

The primary aim of this study was to assess the use of high-resolution multispectral imagery (12 cm/pixel) to estimate the plant water status through different bands and vegetation indexes (VIs) in vines under different pruning management. This work focuses on developing a well-founded indirect method for determining the grapevine water status that could be used to map the Ψ_stem_ in all the vineyard surfaces and aid in irrigation management.

## 2. Materials and Methods 

### 2.1. The Study Vineyard 

This study was carried out during the growing seasons of 2021 and 2022 on a 40 ha commercial vineyard in Yepes (39°56′26.2″ N, 3°42′49.7″ W), Spain, at 699 m above the mean sea level. The vineyard was planted in 2002, cv. Merlot (*Vitis vinifera* L.) over SO4 rootstock (*Vitis berlandieri* × *Vitis riparia*) and arranged on a trellis with a plantation frame of 2.6 m × 1.1 m (3500 vines/ha). The plants have been trained in a double-cordon system. 

The climate of the area corresponds to a typical hot-summer Mediterranean climate. The region has an average daily temperature of 16 °C and an average annual rainfall of 394 mm, mainly concentrated at the end of autumn and the beginning of spring. Summers are characterised by a high atmospheric vapour demand derived from high temperatures (maximum temperatures >40 °C) and low relative humidity. The temperature and relative humidity were measured in-field using a portable weather station OMEGAETTE model HH314A (OMEGA, Ltd., Bridgeport, NJ, USA, EEUU) that provided minute data. From these values, the saturation vapour pressure (e_s_), actual vapour pressure (e_a_) and vapour pressure deficit (VPD) were calculated for the measuring times.

Precipitation and reference evapotranspiration (ETo) data were extracted from the closest public weather station, 18 km from the study site, located in Magán (Toledo) (39°56′06.6″ N 3°56′32.4″ W). The rainfall in 2021 was 347 mm, while the ETo was 1284 mm. In 2022, these values were 348 and 1396 mm, respectively. The meteorological data are summarised in Figure 1. 

### 2.2. Pruning Management 

As in many other large-scale vineyards, this one is divided into smaller units for management purposes, designated as plots or parcels. Our study was executed in one of these plots (Figure 2), belonging to a zone classified as a coarse-loamy, gypsic, mesic Typic Calcixerepts soil [34]. In this 5 ha plot, two pruning strategies have been traditionally used, dividing it into two 2.5 ha areas with homogeneous conditions and management except for pruning. To a large extent, vineyard managers have done so over the last few years to assess its impact on qualitative performance. From this, two pruning treatments were selected for this study. The pruning treatment was developed over 10 years ago. The activity takes place in winter, around the second week of February. This plot area has a more intensive pruning approach. Firstly, mechanical pruning is performed with a horizontal trimmer 30 cm above the cordons to eliminate large amounts of wood in the most cost-effective way. Then, the vines are spur-pruned by hand, leaving approximately two buds per node. The no pruning treatment was developed over 10 years ago. Like the rest of the plots, the activity takes place in winter, around the second week of February. This treatment area was minimally hand-pruned, focusing on removing damaged or unnecessary parts. Therefore, the pruning magnitude is substantially smaller, considering the amount of wood removed after each productive campaign. Six vines out of each pruning system in adjacent lines were selected in order to take the different experimental measurements (Figure 2b).

### 2.3. Irrigation 

Given the atmospheric conditions previously mentioned and the limited irrigation water availability in the growing area, the studied vines inevitably developed under a deficit irrigation strategy. During the irrigation season, from May to August in both years, the ETo was 769 in 2021 and 851 in 2022. Irrigation supplied 17% and 9% of the ETo in 2021 and 2022, respectively. 

Irrigation was applied with one drip emitter per meter with a flow rate of 2 L h^−1^. It was scheduled according to the standard practices followed by Bodegas Casa del Valle, i.e., with limited time intervals. The irrigation periods varied between the years of the study regarding the duration and total quantities applied. In 2021, the irrigation season went from mid-May to the beginning of September, and the irrigation in the study plot was 128 mm. In 2022, the irrigation season was shorter, going from the end of June to the beginning of September, and the applied amount was also reduced, with a total of 82 mm, 36% less than in the previous season. 

### 2.4. Physiological and Agronomic Parameters 

These measurements were conducted in six experimental vines for each pruning treatment in both study campaigns.

#### 2.4.1. Stem Water Potential Measurements 

The Ψ_stem_ (MPa) was measured at 9:00 and 12:00 h solar time on 9 different dates: 25 June 2021, 5 July 2021, 20 July 2021, 30 July 2021, 19 August 2021, 30 June 2022, 15 July 2022, 5 August 2022 and 12 August 2022. Measurements were performed on healthy and shaded leaves from the inner part of the canopy, where 6 leaves/plant were taken per treatment. They were covered with a plastic bag with aluminium foil one hour before the measurement, as standardised methods recommend attaining water status equilibrium between stem and leaves. For this measurement, a Scholander-type pressure chamber was used (Soil Moisture Equipment Corp., Santa Barbara, CA, USA). 

#### 2.4.2. Chlorophyll Measurements 

The leaf chlorophyll level (µmol chlorophyll/m^2^ leaf area) was measured on the same dates and times as the Ψ_stem_ using an Apogee MC-100 sensor (Apogee Instruments Inc., Logan, UT, USA). Measurements were performed on three healthy leaves per experimental vine.

#### 2.4.3. Canopy Description 

Measurements, necessary to describe the canopy development, were taken on 1 July 2021 and 20 June 2022 at an advanced stage in the season, ensuring vegetative growth had halted and maximum vegetative expression had been achieved. 

In each experimental vine, three points were selected to measure the canopy contour (using a flexible tape) and distance between the highest and lowest leaves: trunk and 40 cm on either side. On the same points, the canopy width was noted at three heights (80, 110 and 120 cm from the ground). From these measurements, the canopy height and width were derived, and the canopy volume and external leaf area were calculated:(1)$$Canopy\;volume\left(m^{3}\right)=H*W*SV$$
(2)$$External\;canopy\;area\left(m^{2}\right)=(2H+W)*SV$$
where: H: canopy height (m), W: canopy width (m), SV: spacing between vines (m).

The canopy soil coverage was calculated from a high-resolution (3 cm/pixel) RGB image using QGIS software (version 3.22.13 Białowieża) at the beginning of the irrigation season: 1 July 2021 and 30 June 2022. At this moment, the vegetative development stopped. The apex stops growing and does not develop more leaves (QGIS, Free Software Foundation, Boston, MA, USA). A non-supervised k-means classification with two classes was run to differentiate the vine canopy and soil pixels for a posterior percentage of covered soil calculation (Figure 3).

#### 2.4.4. Quantitative and Qualitative Analysis 

The experimental vines were harvested by hand according to the commercial vineyard manager’s decision concerning the optimal productive characteristics. The selected dates were 20 August 2021 and 16 August 2022. During the harvest, bunches were counted and production was weighed using a portable electronic scale. Between 150 and 200 berries of each vine were randomly selected, packaged, tagged, and stored in a cooler for subsequent analysis.

The selected berries were taken to the laboratory, counted, and weighed immediately. Then, they were processed to determine the total soluble solids (°Brix) with an Atago digital Brix refractometer (ATAGO CO., LTD., Tokyo, Japan), and the pH was measured with a pH Meter Hach sensION (Hach company., Loveland, CO, USA).

### 2.5. Multispectral Images and Vegetation Indexes Calculation 

Aerial images were acquired by employing a UAV, model eBee (AgEagle Aerial Systems Inc., Wichita, Kansas), a commercial fixed-wing platform equipped with a Parrot Sequoia (Parrot© SA, 2017, Paris, France) multispectral sensor. Flights were carried out at 120 m of altitude with nadir mode. On the same days and times as the flights, the Ψ_stem_ measurements were carried out (9:00 and 12:00 solar time). The different high-resolution imagery consisted of RGB images (3 cm/pixel) and multi-band images (12 cm/pixel) that included the green (495–570 nm), red (620–750 nm), red-edge (670–760 nm) and near-infrared (NIR) (780–2500 nm) bands. 

QGIS software was used to extract the band pixel information. Given the high spatial resolution of multi-band images (12 cm/pixel), pure canopy pixels could be selected, avoiding soil interference. A set of five points from the centremost area (top of the canopy) of each vine was selected to obtain each band reflectance value. These reflectance values were loaded on a spreadsheet where the VIs were calculated. Based on the available bands and the cited literature, five indexes (Table 1) were calculated: the NDVI, the Renormalized Difference Vegetation Index (RDVI), the Transformed Chlorophyll Absorption Ratio (TCARI), the Optimised Soil Adjusted Vegetation Index (OSAVI), and the Normalised Difference Red-Edge Index (NDRE).

### 2.6. Statistical Analysis and Stem Water Potential Modelling 

Data subjected to statistical variance analysis (ANOVA) were processed with Infostat Software version 1.5 (Universidad Nacional de Córdoba, Argentina). The mean values were classified using the LSD test (*p* < 0.05). 

In order to estimate the Ψ_stem_ from the UAV-acquired multispectral information, different model approaches were evaluated using Statgraphics 19 software (Statgraphics Technologies, Inc. The Plains, Virginia, USA). Simple linear regressions between the Ψ_stem_ and all individual bands (green, red, red edge and NIR) and simple linear regressions between the Ψ_stem_ and Vis (NDVI, RDVI, TCARI, OSAVI and NDRE) were performed.

Multivariable linear regression models between the Ψ_stem_ and individual bands aim for a better correlation than simple linear regressions.

In the latter case, the Variance Inflation Factor (VIF), which measures multicollinearity, was used for this purpose, excluding solutions where VIF > 10 for any parameter. Moreover, the Durbin–Watson Statistic (DW) was used to assess the autocorrelation between the residuals. Models with DW < 1.5 were rejected. Other metrics we used to select which bands were best to include in the model are the Akaike Information Criterion (AIC), the Schwarz Information Criterion (SBIC), and the Hannan–Quinn Information Criterion (HQC). They are all indicators of the goodness of fit in a multivariable linear regression; the lower the value, the better.

## 3. Results 

### 3.1. Climatic Characterisation 

The climatic variables during the in-field measurements are summarised in Table 2, and the critical monthly parameters are presented in Figure 1. Our results reveal that the 2022 season was hotter (+2.0 °C and +1.8 °C at 9:00 and 12:00, respectively) and drier (10.4% and −3.7% RH lower at 9:00 and 12:00, respectively) than 2021. Moreover, the annual ETo (reference Evapotranspiration) was 109 mm higher in 2022, while a 36% reduced total irrigation was applied. The maximum monthly ETo occurred in July in both years and was 226.1 mm and 249.6 mm in 2021 and 2022, respectively.

The vapour pressure deficit (VPD) suffered an apparent increase between measuring hours due to a decrease in the RH and an increase in the temperature caused by solar radiation. This daily increase reached a topmost value of 98.8% on 5 August 2022. However, the average VPD increase between the hours was higher in 2021 (78.3% vs. 59.5%). Notable differences in climatic conditions between campaigns are also evidenced by the maximum VPD values in 2022, 73.9% and 33.3% higher than in 2022, at 9:00 and 12:00, respectively. These differences assuredly affected the vines’ physiological and spectral behaviour. From our results, it can be noted that the VPD and daily ETo are tightly associated. The peak ETo values coincided with high VPD measures each season, although the ETo is a daily value and the VPD is an averaged punctual value.

### 3.2. Canopy Development 

Characterisation of the different measurements and calculations relative to the canopy structure are reflected in Table 3. As expected, the pruning weight was notably higher in the pruning treatment vines (around double the amount in both years). 

The height, width, volume, and external canopy area within the analysed geometry parameters did not show significant differences between the treatments in the two years of the study. However, the differences between the treatments were found to be influenced more by the effect of the year than by the treatments. On the other hand, the pruning weight and covered soil by the canopy show that the treatment and the year influence significant differences.

Lastly, remote sensing was applied to estimate the percentage of covered soil, which yielded statistically significant observations during both campaigns. This parameter is tightly related to the vine width, as reflected in the results. The pruned vines presented significantly higher soil coverage in both years, and both treatments experimented with an increase in this measure in 2022 (+6% and +9%, pruning and no pruning treatments, respectively).

### 3.3. Physiological Responses 

The plant water status, determined by the Ψ_stem_ measurements, did not show a consistent trend for the different pruning treatments over the two campaigns (Table 4). In 2021, the no pruning treatment expressed lower values of Ψ_stem_ at both times and for almost all the measurement dates, with significant differences between the treatments on three dates and for both measuring times. The average Ψ_stem_ for the 2021 season was significantly lower in the pruning treatment, only at 9:00 solar time. The general tendency observed as the campaign advanced was a reduction in the Ψ_stem_ value, implying a decline in the plant water status for both treatments due to the depletion of water reservoirs in the soil.

In 2022, the physiological behaviour of the pruning treatments reversed. In this campaign, the pruning treatment obtained lower Ψ_stem_ values, with significant differences on three dates and one date for the 9:00 and 12:00 measurements, respectively. The average campaign value was not significant for either of the two times. The progression throughout the season was also a decreasing trend for the Ψ_stem_, for both times and treatments. 

The chlorophyll concentration measurements (Table 5) did not show clear tendencies between the treatments, hours of measurement or campaigns, unlike the Ψ_stem_ values. In 2021, only three single-date significant differences between the pruning treatments were observed at solar noon. In 2022, this number was reduced to one date in the 9:00 measurement. The average campaign values did not show statistical differences for any measuring times.

### 3.4. Vine Production and Quality 

Table 6 summarises the productive and qualitative parameters that were assessed. Only the bunch weight showed significant and consistent differences in both campaigns. The pruning treatment obtained statistically higher values in both years, with an increase of 104% and 34% compared to the no pruning in both years. In 2022, both treatments suffered a decrease in this parameter compared to the previous year: −46% and −19% for pruning and no pruning, respectively. 

The berry weight was higher in the pruning treatment but only significantly so in 2021. The pruned vines presented a higher berry weight than the non-pruned ones but showed the most significant decline between campaigns. The number of bunches per plant behaved contrarily to the previous parameters, as the non-pruned vines had higher numbers during both campaigns, with significant differences found only in the second campaign (+56% and +67% in 2021 and 2022, respectively). 

The production per plant was not significantly different between the pruning treatments, but significant differences were observed between the mean values. In 2021, the pruning treatment had higher production than in 2022, caused by the significantly higher berry and bunch weight. In 2022, the non-pruned plants obtained higher production since the bunch and berry weights were balanced. Moreover, the non-pruned vines developed a very high number of bunches compared to 2021. 

Concerning the qualitative parameters, the TSS (°Brix) did not show significant differences between the treatments in either campaign. the year factor influenced the performance, so in 2022, there was an apparent reduction in this parameter for both treatments, indicating less sugar content in the must. On the other hand, the pH measurements behaved contrarily to the TSS. The 2022 campaign reflected higher pH values for both treatments than the previous one, with significant differences between the treatments, which did not happen in 2021. As with the TSS, with the pH, the year factor influences the quality of the performance.

### 3.5. Vine Spectral Behaviour 

In general terms, the pruning treatment obtained lower values for the NDVI, RDVI, OSAVI, and NDRE in the 2022 season than in 2021 for both flying times (Table 7). In 2021, the indexes average values were statistically higher in the pruning treatment at the 9:00 and 12:00 flights, except for the TCARI. In 2022, there were no significant differences between the two treatments for any of the indexes at both times, except for the NDRE at 9:00. The differences in values between the treatments decreased. In this season, the no pruning treatment reflected higher values for these indexes, although not significantly so (refer to Appendix A Figure A1). The VI values in 2022 were notably more alike between the treatments than in the previous campaign. As a result, statistically significant differences were reduced on individual dates and average seasonal values, as with the Ψstem values (refer to Appendix A Table A1 and Table A2).

The evolution of the different vegetation indexes during the year 2021 for the 9:00 and 12:00 flights is presented in Figure 4. In this year, the two treatments were significantly different for many indexes. The VI values showed a decline as the season advanced, which was more evident in the latest flights, and the same pattern followed in the 2022 season (refer to Appendix A Figure A1). However, the NDVI, RDVI, and OSAVI presented a more evident evolution in time than the rest (Figure 4). The evolution of these indexes shows that in the morning (Figure 4a,c,g), the separation between the treatments is more pronounced than at noon (Figure 4b,d,h).

The NDVI showed no differences in the first two measurement days at 9:00 (Figure 4a). The RDVI and OSAVI showed statistical differences between the treatments for all the dates and times in 2021(refer to Appendix A Table A1 and Table A2). The evolution of the RDVI was constantly equidistant between the two treatments throughout the season at both times (Figure 4c,d). The NDRE did not exhibit a clear difference throughout the season between the two treatments (Figure 4i,j). The differences between the pruning treatments only showed statistical differences for the 9:00 flights in both campaigns (Table 7).

On the other hand, the TCARI presented no clear evolution along the season (Figure 4e,f), where this index was significantly higher in the non-pruned plants at both flying times. The TCARI reversed its response in 2022, and the pruned plants obtained slightly higher average values (Table 7).

A factorial analysis considering the year and treatment (Table 7) showed that the NDVI was influenced by the treatment and year; the RDVI and OSAVI were not influenced by the year but by the treatment. The TCARI was only influenced by the treatment, and the NDRE was influenced by only the year.

Figure 5 shows the NDVI performance between the treatments at two time (9:00 and 12:00) for the last flight date of the 2021 season (19 August). It is observed that the pruning treatment exhibits higher values of the NDVI than the non-pruning treatment at both times of day.

### 3.6. Stem Water Potential Estimation 

#### 3.6.1. Simple Linear Regression Models 

The results of the calculated R^2^ values for the simple linear regressions are depicted in Table 8. Different approaches were developed concerning the data included in the models. Seven different single-variable regression models were performed for each VI value and band reflection (four single-season and single-time, two multi-seasonal and single-time, and one multi-seasonal and multi-time). Even though our objective in this work was to develop a multi-seasonal robust model, the Ψ_stem_ was also predicted for individual seasons and flying times to better understand the results.

In most cases, the correlations between the Ψ_stem_ and the different VIs were stronger at solar midday. The NDVI and OSAVI were, on average, the tightest-fitting VIs (R^2^: 0.55 and 0.51, respectively). Not surprisingly, when modelling the Ψ_stem_ with the combined VI data from 2021 and 2022 or when integrating the measuring times, the R^2^ values decreased due to higher variability being added to the models. 

Concerning the use of individual bands to estimate the Ψ_stem_, the red and green bands obtained the best results (average R^2^: 0.51 and 0.45, respectively) and were the least variable between the models. The red-edge and NIR bands only obtained good correlations in 2021, with exceptionally low values when data from 2022 were included.

Simple linear regression models with the vegetation indexes proved more useful to estimate the Ψ_stem_ when data from both seasons and times were included (*n* = 176) than individual bands, with the NDVI and the red band providing the best fit in each category (R^2^: 0.43 and 0.38, respectively).

#### 3.6.2. Multiple-Variable Regression Modelling with Spectral Bands

A multivariable linear regression model was developed to estimate the Ψstem to improve the quality of the simple regression models (Table 8), especially concerning the multi-seasonal and multi-temporal approach (*n* = 176). Only individual bands were considered independent variables, as the VIs are a linear combination of individual band reflectance values.

The initial procedure included two categorical factors: measurement time and pruning treatment. However, the pruning treatment was not statistically significant at the 95% confidence level. Therefore, it was removed from the categorical factors, and only the measuring time was included. The difference in the average Ψ_stem_ values at 9:00 and 12:00 over the two seasons is statistically significant (−0.92 and −1.32 MPa, respectively), dividing the measuring time into two homogeneous groups to include it as a categorical factor. On the other hand, the Ψ_stem_ values divided by the pruning treatment could not be used to identify two homogeneous groups. 

Of all the possible combinations of bands, the red and red edge yielded the best quality results (highest R^2^ adjusted for degrees of freedom and lowest AIC, HQC, and SBIC). The equation for the fitted model is presented below, and Table 9 summarises the statistical metrics used to evaluate the model performance:(3)Ψstem (MPa)=0.199484∗T+[−0.365892+1.4701∗ RED EDGE−14.8059∗RED]
where:$$\left\{\;\begin{array}{c} T=1\;if\;time\;is\;9:00\;\\ T=-1\;if\;time\;is\;12:00\; \end{array}\right.$$
where *T* is the categorical factor for the time of day the data were obtained. 

This model meets the imposed conditions stated beforehand: Durbin–Watson Statistic >1.5 and VIF for every parameter <10. Therefore, it is accepted as a viable and quality-fitting model. A comparison of the observed and predicted Ψ_stem_ using this model is represented in Figure 6. Noticeably, its coefficient of determination is impressively improved compared to the single-variable regression models. The model, using the red and red-edge bands as independent variables, can explain 72% (R^2^ adj.) of the variance of the dependent variable Ψ_stem_, independently of the pruning management carried out. 

A visual representation of the model for the different measuring times is possible, given that it only has two dependent variables (Figure 7 and Figure 8, for 9:00 and 12:00 solar time, respectively). As observed, lower Ψ_stem_ values are associated with lower red-edge and higher red reflection values. Conceptually, both models are parallel planes separated by the TIME factor, 0.40 MPa. 

## 4. Discussion 

Both years show that climatic conditions in the studied area can be extremely harsh (>40 °C and high evaporative demand). Therefore, optimal water management would be necessary to avoid water stress incidences. The daily ETo values ranged from 6.3 to 9.1 (mm/day) (Table 2). However, it must be noted that the irrigation and rainfall in this commercial vineyard do not fulfil the evaporative demands, and therefore, the vines are grown under a deficit irrigation regime. Both seasons registered similar rainfall (around 348 mm), but in 2021, the annual ETo was lower than in 2022 (1284 vs. 1393 mm), and the irrigation supply was higher (128 vs. 82 mm).

Our results indicate that the vine water status presented a wide range of values in both seasons. In 2021, the measured solar midday Ψ_stem_ values oscillated between −0.8 MPa (beginning of the measuring campaign) and −1.6 MPa (last measurement day). According to the thresholds described by [40], these values correspond to non-stressed vines and intense water stress, respectively. In 2022, the solar midday Ψ_stem_ readings were noticeably lower, ranging from −0.9 MPa to −1.9 MPa, which can be interpreted as non-stressed and severely stressed vines, respectively.

Vine water use is majorly determined by atmospheric conditions (VPD) and plant structural characteristics like the canopy size or disposition. The VPD is an integrating climatic parameter more tightly associated with physiological variables such as the stomatal conductance or Ψ_stem_. Ref. [41] concluded that around 71% of the variability in the Ψ_stem_ could be explained by the VPD, and this correlation was not affected by the location or cultivar. Given that the climatic and irrigation conditions were equal for both pruning treatments, it could be assumed that the differences in their water status were caused almost solely by canopy management. However, climatic data can be used to explain the general behaviour of the crop during the day and throughout the campaign. The atmospheric conditions were clearly reflected in the Ψ_stem_ values, evidencing their intrinsic relationship. The 2022 campaign was subject to higher evaporative demands (Table 2), resulting in lower Ψ_stem_ values and vine water status. The minimal Ψ_stem_ values in 2022 were 54.5 and 18.8% lower than in 2021 for the 9:00 and 12:00 measures, respectively. It is also remarkable that the lowest value at 9:00 in 2022 (−1.7 MPa) was even smaller than the lowest 12:00 value in 2021 (−1.6 MPa).

The results from both campaigns indicate that the Ψ_stem_ decreases throughout the day due to transpiration, obtaining lower values in almost all cases in the midday measure for the same treatment (Table 4). This tendency has been widely observed and studied [9]. In our study, the midday value was never higher than the morning one, only equal in one event (15 July 2022, pruning treatment). The midday Ψ_stem_ reading was 58.3% and 12.9% lower than at 9:00 in 2021 and 2022, respectively. Considering both seasons, the average Ψ_stem_ values significantly differed between the measuring times: −0.92 and −1.32 MPa at 9:00 and 12:00, respectively. This indicates that the Ψ_stem_ reduction during the day is less notable under more stressed conditions, as with the VPD (Table 2). The average VPD values were 2.7 and 4.5 kPa at 9:00 and 12:00, respectively. This is also evidenced by the fact that in both campaigns, the treatment with the lowest average Ψ_stem_ values presented the most negligible perceptual differences between the measuring times (no pruning in 2021, pruning in 2022).

The effect of pruning severity on canopy development (Table 3) did reveal some interesting differences that were not statistically significant in most cases. Ref. [42] concluded that less intensive pruning approaches resulted in higher nodes and shoots per vine but shorter shoot lengths. This can explain our results: the pruning treatment vines grew longer shoots, which developed more laterally, resulting in a higher percentage of soil coverage, although it has not been reflected in the rest of the parameters of the geometry of the vine, which did not show significant differences. These results suggest that remote-sensing tools may be more accurate in determining the breathable surface of plants than field measurements. Remote sensing has used airborne imagery to map the relative differences in vine canopies, which are used to characterise the grapevine canopy shape and vegetative expression throughout a vineyard [32]. 

Concerning the effect of water stress on vine production and berry quality (Table 6), our results are in agreement with other evaluated studies. Like [6], we found that more heavily water-stressed vines (no pruning treatment in 2021; pruning treatment in 2022) reported a lower yield per plant and smaller berries in both study seasons. The same results were observed by [26], who also reported that irrigated vines maintained at Ψ_stem_ values above −1.0 MPa until harvest produced higher berry flesh mass and had a lower skin to pulp relation. Regarding the fact that the no pruning treatment had a higher yield in 2022 compared to the pruning treatment, it could be explained by a higher precipitation in the spring of 2022 compared to 2021 (Figure 1). This could have influenced the sprouting of buds, greater number of bunches and then the bunch weight [1]

In our study, the Ψ_stem_ variation was not caused by irrigation management but by pruning management; however, the physiological response of the vines regarding production was the same. 

Many studies have addressed the relationship between the VIs and healthy vegetative growth of crops. Our study observed that the VIs exhibited higher values between the two pruning treatments (Figure 4). However, when extreme climatic conditions occur, as in the 2022 season, plant stress increases. Such conditions affected the behaviour of the VIs and not all of them could detect the water stress experienced by the vines [27]. This stress, however, was detected by the Ψ_stem_ measurement. 

The 2022 conditions limited the performance in the VIs. Severe water deficits decrease the leaf area and intercepted light [28] but also induce stomatal closure, which limits photosynthesis [25]. In addition, in grapevines, changes in the leaf pigment composition have been associated with water stress [43,44,45]. This situation makes it difficult to detect differences between treatments. However, another factor to consider in the evolution of the indexes is the angle of illumination from the sun, as it is not the same at the beginning of summer as at the end.

The central part of our study, the prediction of the vine Ψ_stem_ using high-resolution (12 cm/pixel) multispectral sensors mounted on UAVs, proved to be a successful tool in vineyard water status variability management. However, in most cases, the prediction of the Ψ_stem_ was stronger at solar noon than at 9:00 (Table 8), possibly due to the better quality spectral data derived from a more vertical sun position. 

These values show that the NDVI is the one most strongly correlated with vine water status. However, indexes such as the OSAVI and RDVI, which have shown significant differences between the treatments in conditions where the Ψ_stem_ did not exhibit differences, suggest that they are related to more structural aspects than physiological ones of the crop. This would explain why both the OSAVI and RDVI were not influenced by the year (Table 7). Indexes integrating the red and NIR bands are often more associated with the structural characteristics of plants, such as the vigour, biomass, leaf area, etc. [32,46]. The NDVI includes these bands, although its use has also been linked to canopy vigour [32]; in our study, it was also able to correlate with water status. Other authors have found that the NDVI can be a good indicator of the plant water status [21,47]. 

The NDVI may be more related to the amount of intercepted light than to the leaf area or biomass. Ref. [43] observed that the Ψp vs. NDVI relationship was consistent across their study, vegetative development was strongly determined by water availability, and the vegetation index NDVI effectively characterised the effects of water availability on vine canopy growth. 

Other authors did not find a relationship between the NDVI and water status under severe stress conditions. Reflectance indexes such as the NDVI or the simple ratio (SR) are useful for characterising the canopy structure and pigment concentration, and, thus, the potential photosynthetic activity [43], but they have proven less useful for monitoring photosynthetic functioning under stress conditions [48,49]. Thus, these indexes have proven less useful for monitoring plant physiological status (e.g., photosynthesis and/or water status) under stress conditions [43].

Different regression model approaches yielded different results (Table 8). The single-season and single-time simple Ψ_stem_ regressions were strongly associated with the calculated Vis. For this model type, the correlations were quite variable (R^2^ range: 0.12–0.73), and the OSAVI index achieved the best fit (R^2^ = 0.73) for the average 9:00 measure in 2021. Ref. [24] estimated the Ψ_stem_ for a single day with a high-resolution multispectral sensor (10 cm/pixel) multiple camera array (MCA-6, Tetracam, Inc., California, USA) on board a UAV but obtained an R^2^ value of only 0.35 when using the OSAVI. However, our results were similar to theirs with respect to the capacity of the NDVI and TCARI for estimating the Ψ_stem_. They obtained a significant fit for both indexes, with R^2^ values of 0.68 and 0.45 (NDVI and TCARI, respectively). In our case, the TCARI expressed the highest correlation in the 9:00 multi-seasonal model (R^2^ = 0.56, *n* = 88) and the NDVI at 12:00 for the same type of model (R^2^ = 0.58, *n* = 88). When combining data from both flying times, the NDVI was once again the best index for predicting the Ψ_stem_ (R^2^ = 0.43, *n* = 176).

Additionally, ref. [5] could relate the NDVI and stomatal conductance on a single day with a correlation of r = 0.56, revealing that VIs can predict different vine water status indicators. Proximal-sensing methodologies have also proven useful in correlating VIs with the Ψ_stem_. For example, ref. [50] reported a significant linear relationship between the NDVI and Ψ_stem_ (R^2^ = 0.69) when using a hand-held spectrometer at canopy level, which is fundamentally similar to a UAV-mounted sensor.

Contrarily to other studies [13,14,51], we did not obtain satisfactory findings from the chlorophyll concentration measurements (Table 5), nor were we able to correlate it with the VIs. This could be because their studies used more precise destructive methods for determining the leaf chlorophyll concentration levels.

Normally, there were more works relating the Ψ_stem_ with several VIs than with spectral bands. Recently, ref. [52] showed that using spectral bands as Ψ_stem_ predictors provided better results than VIs in olive orchards, using machine-learning techniques. Our results revealed some interesting features of this approach. In some cases (2022, 9:00 and 12:00; 2021 and 2022, 9:00 and 12:00) (Table 8), the R^2^ values were higher when using individual bands than VIs. The red band correlated strongly with the Ψ_stem,_ obtaining an R^2^ as high as 0.71. On the other hand, a downside of using individual bands is that the variation in the R^2^ values was higher than with the VIs: the NDVI ranged from 0.5 to 0.65, while the red band ranged from 0.02 to 0.71.

To our mind, the most significant achievement of this work has been the development of a robust multi-seasonal linear model based on the red and red-edge bands, which predicts the Ψ_stem_ better than any of the tested VIs (Table 9). Further validation is needed to ensure its applicability using linear and non-linear methods. However, in the first instance, it does seem like a very valid method for detecting spatial and temporal water status variability in vineyards.

## 5. Conclusions

Using high-resolution (12 cm/pixel) multispectral imagery acquired by UAV-mounted sensors has proven useful in predicting the vine water status (Ψ_stem_) in a semi-arid commercial vineyard in central Spain. Other authors have already found it useful in vineyards [19,20]. Our findings have revealed that simple or multiple-regression models using individual bands reflectance values and vegetation indexes yield significant correlations with the Ψstem independently of canopy development. The multispectral image acquisition time is an important factor to consider, given that the solar midday flights obtained better fittings than at 9:00. Nonetheless, by including the time as a categorical factor in our multivariable model, its effect was reduced, and a robust model was achieved. This multivariable model (R^2^ = 0.74) with the red (620–750 nm) and red-edge (670–760 nm) bands could only be used to predict the Ψstem. Although other authors have found that the NDVI can be a good indicator of the plant water status [21,50]. In our study, the relation of the Ψstem with the NDVI is lower (R^2^ = 0.65). This results in us being able to use cheaper custom-made sensors with just the two necessary bands.

The most significant achievement of this work has been the development of a robust multi-seasonal linear model based on the red and red-edge bands, which predicts the Ψstem better than any of the tested VIs. This research helps farmers to determine different irrigation management zones.

## 6. Future Research

This work opens up the possibility for future research focused on different aspects. As a future prospective, this study should be applied to other vineyards with different cultivars to validate the model. Exploration of the integration of AI and machine-learning methods should focus on automated data analysis, potentially enhancing the accuracy and enabling real-time application in diverse settings.

## Figures and Tables

**Figure 1 plants-13-01350-f001:**
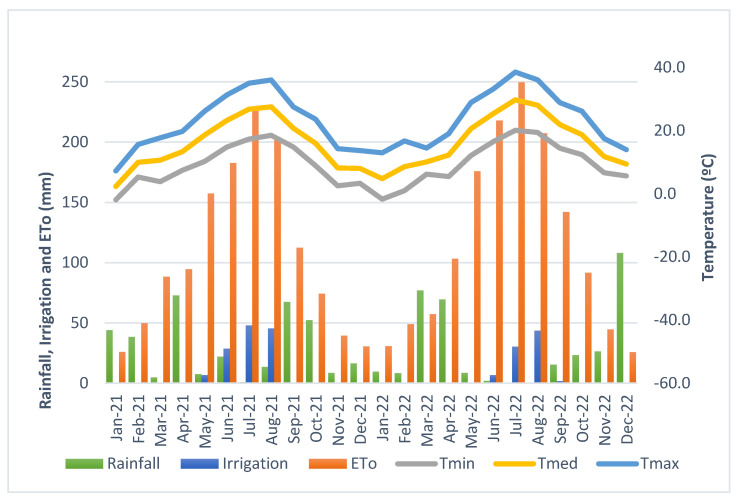
Monthly climatic variables during 2021 and 2022: rainfall, applied irrigation, reference evapotranspiration (ETo), average minimum temperature (Tmin), average temperature (Tmed) and average maximum temperature (Tmax). Data were obtained from the Magán public weather station (39°56′06.6″ N 3°56′32.4″ W).

**Figure 2 plants-13-01350-f002:**
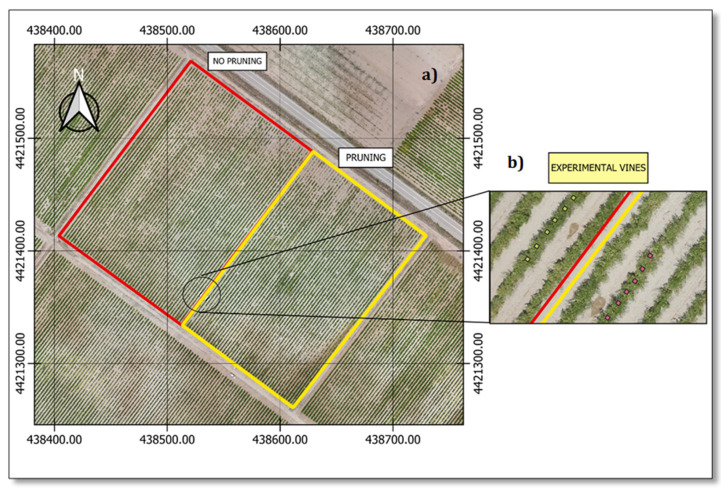
(**a**) Aerial RGB image of the commercial vineyard used for this study. Different pruning management plots (2.5 ha each) are outlined in different colours: red for no pruning and yellow for pruning. (**b**) Location details of the six experimental vines where measurements were taken.

**Figure 3 plants-13-01350-f003:**
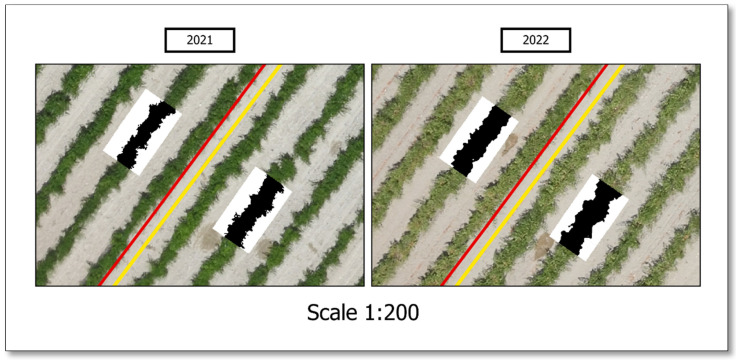
Details of the non-supervised classification performed to calculate the percentage of covered soil of the two pruning treatments in two consecutive seasons—a commercial vineyard in Yepes (Toledo).

**Figure 4 plants-13-01350-f004:**
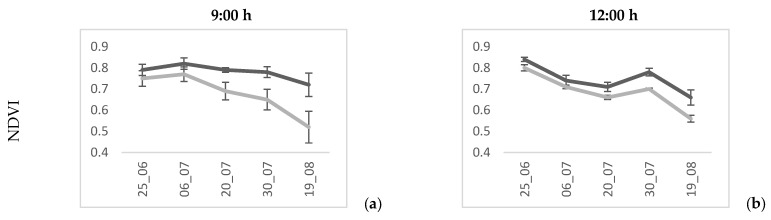
Evolution along the season of different vegetation indexes calculated from a high-resolution (12 cm/pixel) multispectral sensor mounted on board a UAV at 09:00 and 12:00 solar time in 2021 on a commercial vineyard under two pruning treatments (mechanical and no pruning) in central Spain (Yepes, Toledo).

**Figure 5 plants-13-01350-f005:**
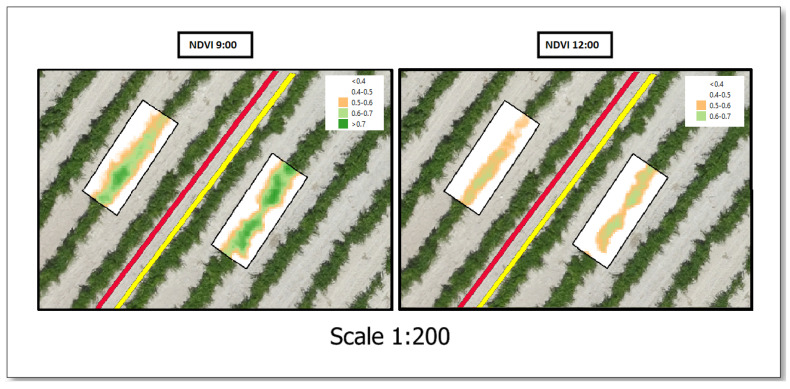
Details of the NDVI of two pruning treatments at two times of day. Left imagen corresponding to values of the NDVI at 9:00. Right imagen corresponding to values of the NDVI at 12:00. The no pruning treatment is located near the red line; the pruning treatment is located near the yellow line.

**Figure 6 plants-13-01350-f006:**
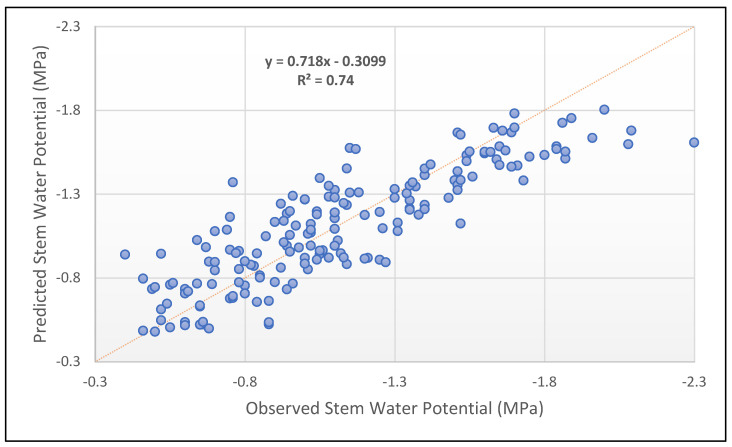
Plot of the observed vs. predicted Ψ_stem_ using the developed multivariable regression model with the red and red-edge bands as independent variables (*n* = 176). The dashed line indicates a 1:1 slope.

**Figure 7 plants-13-01350-f007:**
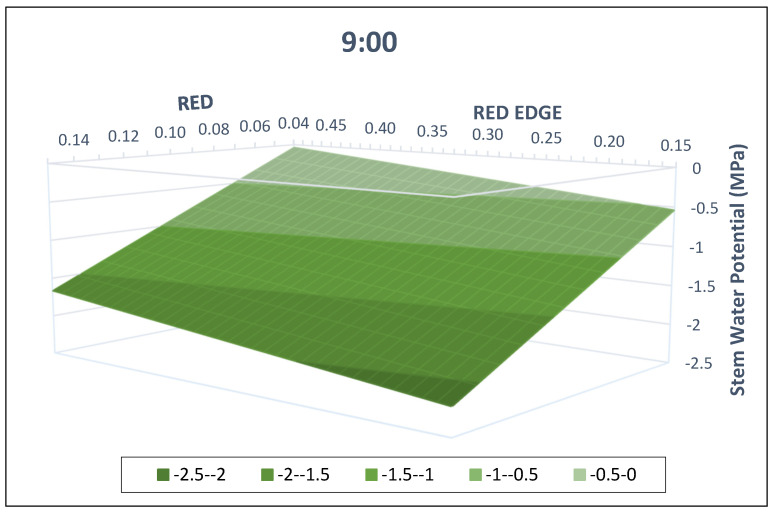
Visual representation of the multivariable linear regression model developed at 9:00. Horizontal axes correspond to the red and red-edge reflection values, and the vertical axis to the stem water potential (MPa). The model developed includes data from 2021 and 2022 and two pruning treatments for commercial vineyards in Yepes (Spain).

**Figure 8 plants-13-01350-f008:**
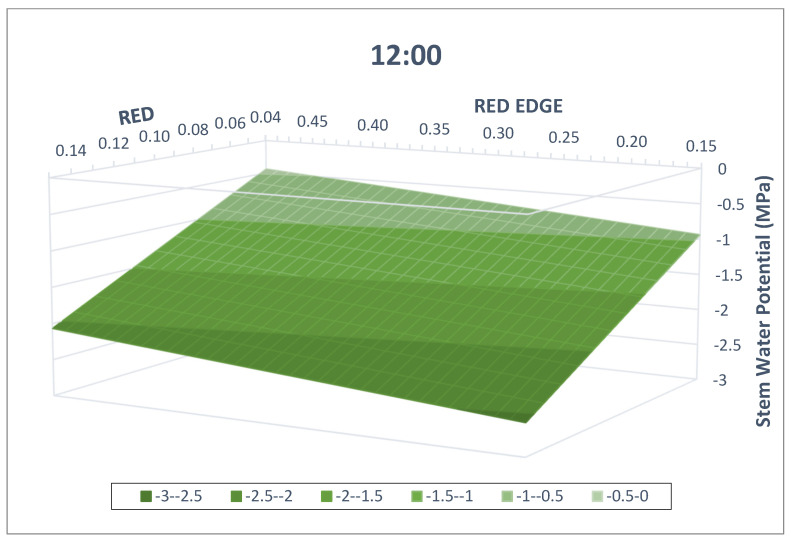
Visual representation of the multivariable linear regression model developed at 12:00. Horizontal axes correspond to the red and red-edge reflection values, and the vertical axis to the stem water potential (MPa). The model developed includes data from 2021 and 2022 and two pruning treatments commercial vineyard in Yepes (Spain).

**Table 1 plants-13-01350-t001:** Formulas and references of the vegetation indexes used in this study.

Index	Formula	Reference
NDVI	$NDVI=\frac{R_{NIR}-R_{red}}{R_{NIR}+R_{red}}$	[35]
RDVI	$RDVI=\frac{R_{NIR}-R_{red}}{\sqrt{R_{NIR}+R_{red}}}$	[36]
TCARI	$TCARI=3*[\left(R_{RedEdge}-R_{red}\right)-0.2*\left(R_{RedEdge}-R_{green}\right)*\left(\frac{R_{RedEdge}}{R_{red}}\right)]$	[37]
OSAVI	$OSAVI=(1+0.16)*\left(R_{NIR}-R_{red}\right)/\left(R_{NIR}+R_{red}+0.16\right)$	[38]
NDRE	$NDRE=\frac{R_{NIR}-R_{Red\;Edge}}{R_{NIR}+R_{Red\;Edge}}$	[39]

**Table 2 plants-13-01350-t002:** Temperature (°C), relative humidity (%), vapour pressure deficit (kPa) and daily ETo (mm/day) during the times and days that the Ψ_stem_ was evaluated and multispectral images were obtained for two hours of measurement and two campaigns in a commercial vineyard under two pruning treatments (mechanical and no pruning) in central Spain (Yepes, Toledo).

	9:00 Solar Time			12:00 Solar Time			ETo (mm/Day)
Date	T (°C)	RH (%)	VPD (kPa)	T (°C)	RH (%)	VPD (kPa)	
25 June 2021	27.4	43.9	2.05	33.5	29.8	3.63	6.6
5 July 2021	28.2	42.8	2.19	34.0	30.8	3.68	8.2
20 July 2021	28.8	39.6	2.39	35.9	19.8	4.74	7.9
30 July 2021	28.3	34.1	2.53	33.6	12.7	4.54	9.1
19 August 2021	28.7	41.0	2.32	33.1	22.8	3.91	6.3
Average 2021	28.3	40.3	2.30	34.0	23.2	4.10	7.6
30 June 2022	27.5	28.2	2.64	30.0	23.1	3.25	7.0
15 July 2022	35.2	22.6	4.40	40.2	15.4	6.32	8.1
5 August 2022	29.1	37.7	2.52	36.6	18.4	5.01	7.3
12 August 2022	29.3	31.0	2.81	36.6	21.2	4.85	6.3
Average 2022	30.3	29.8	3.09	35.8	19.5	4.86	7.2

**Table 3 plants-13-01350-t003:** Vine canopy geometric parameters relative to the development and architecture over two campaigns (2021 and 2022), under mechanical pruning (P) and no pruning (NP), in a commercial vineyard in central Spain (Yepes, Toledo). Mean data and coefficient of variation (CV).

	2021					2022				
	P	CV (%)	NP	CV (%)		P	CV (%)	NP	CV (%)	
Pruning weight (g/plant)	177.5	27%	87.5	29%	*	216.9	30%	101.7	35%	*
Canopy height (cm)	69.75	21%	70.67	18%	ns	86.25	25%	106.08	11%	ns
Width (cm)	71.52	19%	58.9	15%	ns	96.07	10%	89.73	19%	ns
External canopy area (m^2^)	2.34	8%	2.07	12%	ns	3.06	10%	3.14	9%	ns
Canopy volume (m^3^)	0.53	13%	0.46	23%	ns	0.91	27%	1.04	14%	ns
Canopy contour (m)	1.64	11%	1.64	14%	ns	1.72	15%	1.73	6%	ns
Covered soil by the canopy (%)	43%	6%	36%	9%	*	49%	4%	45%	2%	*

ns: non-significant and * significant at *p* < 0.05.

**Table 4 plants-13-01350-t004:** Ψ_stem_ (MPa) measured in two hours per day and over two campaigns (2021 and 2022), under mechanical pruning (P) and no pruning (NP), in a commercial vineyard in central Spain (Yepes, Toledo).

	9:00 Solar Time			12:00 Solar Time		
Date	P Ψ_stem_ (MPa)	NP Ψ_stem_ (MPa)		P Ψ_stem_ (MPa)	NP Ψ_stem_ (MPa)	
25 June 2021	−0.5	−0.6	ns	−0.8	−0.8	ns
5 July 2021	−0.6	−0.7	ns	−1.1	−0.9	ns
20 July 2021	−0.5	−0.8	*	−1.0	−1.1	*
30 July 2021	−0.6	−0.7	*	−1.0	−1.4	*
19 August 2021	−0.7	−1.1	*	−1.3	−1.6	*
Average 2021	−0.6	−0.8	*	−1.0	−1.2	ns
30 June 2022	−0.9	−0.9	ns	−1.1	−1.1	ns
15 July 2022	−1.1	−0.8	*	−1.1	−0.9	*
5 August 2022	−1.7	−1.5	*	−1.9	−1.7	ns
12 August 2022	−1.6	−1.3	*	−1.7	−1.7	ns
Average 2022	−1.3	−1.1	ns	−1.4	−1.3	ns

ns: non-significant and * significant at *p* < 0.05.

**Table 5 plants-13-01350-t005:** Chlorophyll concentration measurements (µmol chlorophyll/m^2^ leaf area), two hours per day and over two campaigns (2021 and 2022), under mechanical pruning (P) and no pruning (NP), in a commercial vineyard in central Spain (Yepes, Toledo).

	9:00 Solar Time			12:00 Solar Time		
Date	P Chlorophyll µmol Chlorophyll/m^2^ Leaf Area	NP Chlorophyll µmol Chlorophyll/m^2^ Leaf Area		P Chlorophyll µmol Chlorophyll/m^2^ Leaf Area	NP Chlorophyll µmol Chlorophyll/m^2^ Leaf Area	
25 June 2021	15.00	17.05	ns	14.05	15.65	ns
5 July 2021	16.05	17.18	ns	19.75	14.65	*
20 July 2021	17.18	15.80	ns	20.40	17.08	ns
30 July 2021	14.45	14.75	ns	18.23	15.38	*
19 August 2021	18.98	16.32	ns	18.45	15.61	*
Average 2021	16.33	16.22	ns	18.18	15.67	ns
30 June 2022	14.9	14.7	ns	14.5	14.9	ns
15 July 2022	16.2	14.9	*	16.7	15.8	ns
5 August 2022	15.8	15.0	ns	15.6	15.5	ns
12 August 2022	14.1	14.2	ns	14.7	14.7	ns
Average 2022	15.3	14.7	ns	15.4	15.2	ns

ns: non-significant and * significant at *p* < 0.05.

**Table 6 plants-13-01350-t006:** Productive and qualitative parameters were evaluated for two campaigns (2021 and 2022), under mechanical pruning (P) and no pruning (NP), in a commercial vineyard in central Spain (Yepes, Toledo).

	2021			2022		
	P	NP		P	NP	
Production per plant (kg)	1.88	1.53	ns	1.27	1.60	ns
Berry weight (g)	0.77	0.51	*	0.39	0.34	ns
Bunch weight (g)	52.2	25.6	*	27.96	20.86	*
Number of bunches per plant	36.25	56.5	ns	37.17	62.0	*
TSS (°Brix)	28.48	28.88	ns	25.8	25.4	ns
pH	3.44	3.41	ns	3.65	3.81	*

ns: non-significant and * significant at *p* < 0.05.

**Table 7 plants-13-01350-t007:** Annual average and factorial analysis (FA) of vegetation indexes calculated from a high-resolution (12 cm/pixel) multispectral sensor mounted on board a UAV at two solar times (09:00 and 12:00), for two campaigns (2021 and 2022), under mechanical pruning (P) and no pruning (NP), in a commercial vineyard in central Spain (Yepes, Toledo).

9:00		NDVI		RDVI		TCARI			OSAVI		NDRE	
2021	P	0.78	*	0.61	*	0.03	*		0.71	*	0.23	*
	NP	0.67		0.47		0.17			0.58		0.21	
2022	P	0.67	ns	0.56	ns	0.29	ns		0.63	ns	0.18	*
	NP	0.70		0.56		0.23			0.65		0.19	
12:00												
2021	P	0.74	*	0.55	*	0.06	*		0.67	*	0.20	ns
	NP	0.69		0.47		0.15			0.59		0.20	
2022	P	0.68	ns	0.51	ns	0.18	ns		0.61	ns	0.19	ns
	NP	0.71		0.51		0.15			0.63		0.20	
FA												
Treatment	P NP	*		*		ns		*			ns	
Year	2021 2022	*		ns		*		ns			*	

ns: non-significant and * significant at *p* < 0.05.

**Table 8 plants-13-01350-t008:** Coefficients of determination (R^2^) for the simple linear regression models where the Ψ_stem_ is predicted using the five calculated VIs (NDVI, RDVI, TCARI, OSAVI and NDRE) and the four individual bands (red, green, red edge and NIR) in different model-type scenarios.

Model Type	Single-Season and Single-Time				Multi-Season 2021–2022		
	2021 *		2022 **		Single Time ***		Both Times
Indexes	9:00	12:00	9:00	12:00	9:00	12:00	9:00–12:00
NDVI	0.5	0.65	0.62	0.65	0.44	0.58	0.43
RDVI	0.49	0.6	0.37	0.48	0.12	0.39	0.23
TCARI	0.13	0.40	0.53	0.56	0.56	0.48	0.36
OSAVI	0.73	0.65	0.52	0.58	0.25	0.49	0.33
NDRE	0.23	0.41	0.30	0.12	0.40	0.00	0.10
**Bands**							
RED	0.02	0.47	0.71	0.68	0.69	0.64	0.38
GREEN	0.27	0.21	0.62	0.72	0.50	0.62	0.18
RED EDGE	0.51	0.25	0.0008	0.12	0.03	0.08	0.00
NIR	0.46	0.42	0.03	0.02	0.00	0.08	0.04

* *n* = 40, ** *n* = 48, *** *n* = 88.

**Table 9 plants-13-01350-t009:** Performance statistical metrics of the multivariable linear regression model used to predict the Ψ_stem_ with data from 2021 and 2022 and two flying times, 9:00 and 12:00 solar time: *n*, R^2^ and R^2^ adjusted for degrees of freedom (R^2^ adj.), standard error of estimates (S_e_), mean square error (MSE), mean absolute error (MAE) and Durbin–Watson Statistic (DW).

Metric	Value
*n*	176
R^2^	0.72
R^2^ adj.	0.72
S_e_	0.215
MSE	0.05
MAE	0.17
DW	1.53

## Data Availability

Data are contained within the article.

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
