# Peer review of "Application of Unmanned Aerial Vehicle (UAV) Sensing for Water Status Estimation in Vineyards under Different Pruning Strategies"

_plants, 2024, doi:10.3390/plants13101350_

Round 1
Reviewer 1 Report
Comments and Suggestions for Authors
Reviewer
This research paper represents pruning determines plant water status due to its effects on leaf area and thus irrigation 10 management. The primary aim of research article was to assess the use of high-resolution multispectral 11 imagery to estimate plant water status through different bands and Vegetation Indexes (VI) and to 12 evaluate which is most suitable under different pruning management. This work was carried out in 13 2021 and 2022 in a commercial Merlot vineyard in an arid area in central Spain. Two different pruning strategies were carried out: mechanical pruning and no pruning. Stem water potential was meas-15 ured with a pressure chamber (Ψstem) at two different solar times (9h and 12h). I have some questions about this paper.
Line: 29, to fix reductio n to reduction.
Line: 85, edit the sentence appropriately, too many parentheses/parentheses in the sentence.
Line: 113, correct unit, instead 0.12 m/pixel write 12 cm/pixel. Unify this unit throughout the research paper.
After chapter 1. Introduction comes chapter 2. Material and Methods and not chapter 2. Results ??? Please insert chapter 2. Material and Methods instead of chapter 2. Results.
Line 122: is the correctly written the value RH -10.4% ??? Relative humidity is measured from 0% to 100%.
Line 123: in the text specify more detailed the meaning of label ETo.
Line 127: in the text specify more detailed the meaning of label VPD.
Please correct the table 3 in the research paper, the parameters in the table are not displayed in an understandable way, where in the table 3 is the Stem parameter located ???
Please correct the table 4 in the research paper, the parameters in the table are not displayed in an understandable way, where in the table 4 is the Chlorophyll concentration parameter located ???
Comment on the results in table 5, namely why the yield per vine was higher in 2022 "no pruning method".
Line 230-231: Figure A1. Refer to the appendix.
Line 233: Table A1 and Table A2. Refer to the appendix.
Please correct the table 6 in the article, the parameters in the table are not displayed in an understandable way.
Line 245: Figure 2 A, C, G or better Figure 2 a, c, g ???
Line 246: Figure 2 B, D, H or better Figure 2 b, d, h ???
Line 253: Table A1 and A2. Refer to the appendix.
Line 254: Figure 4 C, D or Figure 2 c, d ???
Line 255: Figure 4 J, K or Figure 2 j, k ???
Line 257: Table 6 or another Table x ???
Line 258: Figure 4 E, F or Figure 2 e, f ???
Please correct the table 7 in the research paper, the parameters in the table are not displayed in an understandable way.
Line 310: the equation label is missing.
Line 396-406: contents belong to chapter 1. Introduction.
Line 458: which high resolution sensor, which type of sensor and sensor producer?
Replace Chapter 4. Material and Methods with Chapter 2.
Line 515: correct subchapter 5.2 title, instead Prinung management to write Pruning management!
Line 515: did you correctly label Figure 1 in the text???
Line 531: did you correctly label Figure 1b in the text???
Line 533: did you correctly label Figure 1 in the text???
Line 575-576: the equation label is missing.
Line 586: correct the label for Figure 2. twice in the text you mention two same tags for different images???
Line 638: correct unit, instead 0.12 m/pixel write 12 cm/pixel. Unify this unit throughout the research paper.
Revise the entire Results chapter and describe the results of your research work more scientifically. Chapter Results sound too technically.
Revise chapter 5. Conclusions. You wrote down the conclusions of your research work very modestly. give a more detailed scientific description of your research work and compare it with other research works.
Add chapter 6. Future work in the research paper.

Author Response
Dear reviewer
Thanks for the comments on the document.
The modifications have been highlighted in the manuscript.
I enclose the response to the comments
Kind regards
Line: 29, to fix reductio n to reduction.
The writing error has been corrected in line 29
Line: 85, edit the sentence appropriately, too many parentheses/parentheses in the sentence.
Excess parentheses have been reduced in line 85
Line: 113, correct unit, instead 0.12 m/pixel write 12 cm/pixel. Unify this unit throughout the research paper.
The units have been corrected in line 124
After chapter 1. Introduction comes chapter 2. Material and Methods and not chapter 2. Results ??? Please insert chapter 2. Material and Methods instead of chapter 2. Results.
Thank you for your recommendation, we change the order that you suggested
Line 122: is the correctly written the value RH -10.4% ??? Relative humidity is measured from 0% to 100%.
It was a writing error, it was trying to explain that the humidity observed in 2022 was lower than that observed in 2021. The new sentence is in lines 287-289 “Our results manifest that the 2022 season was hotter (+2.0ºC and +1.8ºC at 9:00 and 12:00, respectively) and drier (10.4% and -3.7% RH lower at 9:00 and 12:00, respectively) than 2021”
Line 123: in the text specify more detailed the meaning of label ETo.
The label of ETo has been written, line 289 “, annual ETo (reference Evapotranspiration)”
Line 127: in the text specify more detailed the meaning of label VPD.
The label of VPD has been written, line 293 “Vapor Pressure Deficit (VPD)”
Please correct the table 3 in the research paper, the parameters in the table are not displayed in an understandable way, where in the table 3 is the Stem parameter located ???
The stem water potential parameter of table 3 (now renamed Table 4) has been written, line 344
Please correct the table 4 in the research paper, the parameters in the table are not displayed in an understandable way, where in the table 4 is the Chlorophyll concentration parameter located ???
The chlorophyll parameter of table 4 (now renamed Table 5) has been written, line 353
Comment on the results in table 5, namely why the yield per vine was higher in 2022 "no pruning method".
The explanation of this condition is located in lines 580-583
Line 230-231: Figure A1. Refer to the appendix.
The Figure A1 has been corrected
Line 233: Table A1 and Table A2. Refer to the appendix.
Tables A1 and A2 have been corrected
Please correct Table 6 in the article, the parameters in the table are not displayed in an understandable way.
Table 6 (now renamed Table 7) has been modified for a better understanding. Line 401
Line 245: Figure 2 A, C, G or better Figure 2 a, c, g ???
We changed the capital latter
Line 246: Figure 2 B, D, H or better Figure 2 b, d, h ???
We changed the capital latter
Line 253: Table A1 and A2. Refer to the appendix.
Tables A1 and A2 have been corrected in line 422
Line 254: Figure 4 C, D or Figure 2 c, d ???
We changed the capital latter
Line 255: Figure 4 J, K or Figure 2 j, k ???
We changed the capital latter
Line 257: Table 6 or another Table x ???
We changed the capital latter
Line 258: Figure 4 E, F or Figure 2 e, f ???
We changed the capital latter
Please correct the table 7 in the research paper, the parameters in the table are not displayed in an understandable way.
Table 7 (now renamed Table 8) has been modified for a better understanding. Line 465
Line 310: the equation label is missing.
The equation labels have been added in line 485
Line 396-406: contents belong to chapter 1. Introduction.
We changed the content to chapter one. Lines 95 to 100
Line 458: which high-resolution sensor, which type of sensor and sensor producer?
We added the sensor characteristics in lines 631-632 “with high-resolution multispectral sensor (10cm/pixel) Multiple Camera Array (MCA-6, Tetracam, Inc., California, USA) onboard UAV”
Replace Chapter 4. Material and Methods with Chapter 2.
we changed the order that you suggested
Line 515: correct subchapter 5.2 title, instead Prinung management to write Pruning management!
We corrected the writing error
Line 515: did you correctly label Figure 1 in the text???
We corrected the label error. Figure 1 is meteorological information along the text
Line 531: did you correctly label Figure 1b in the text???
We corrected the label error. Figure 1b has been removed
Line 533: did you correctly label Figure 1 in the text???
We corrected the label Figure 1 in the text
Line 575-576: the equation label is missing.
The equation labels have been added in lines 219-220
Line 586: correct the label for Figure 2. twice in the text you mention two same tags for different images???
The error has been corrected
Line 638: correct unit, instead 0.12 m/pixel write 12 cm/pixel. Unify this unit throughout the research paper.
The units have been corrected
Revise the entire Results chapter and describe the results of your research work more scientifically. Chapter Results sound too technically.
Thank you for the comment. We have rewritten some parts of the result to be more scientific.
Revise chapter 5. Conclusions. You wrote down the conclusions of your research work very modestly. give a more detailed scientific description of your research work and compare it with other research works.
We have rewritten the conclusions with more scientific details and adding references to other authors. Lines 676 to 683 “Although other authors have found that NDVI can be a good indicator of plant water sta-tus[21,45]. In our study the relation of Ψstem with NDVI is lower (R2= 0.65). This results in being able to use cheaper custom-made sensors with just the two necessary bands.
The most significant achievement of this work has been the development of a robust multi-seasonal linear model based on the RED and RED EDGE bands, which predicts Ψstem better than any of the tested VIs. This research helps farmers to determine different irrigation management zones”
Add chapter 6. Future work in the research paper.
We added the new chapter in lines 684 to 689 “This work open the possibility for future researches focused on different aspects. as a future prospective, this study should be applied to other vineyards with different cultivars to validate the model. Exploration of the integration of AI and machine learning methods focus to automate data analysis, potentially enhancing accuracy and enabling real-time application in diverse settings”
Reviewer 2 Report
Comments and Suggestions for Authors
A wonderful article dedicated to the use of multispectral images to control plant parameters. However, a number of changes should be made to improve the quality of the presentation of the results.
1. There are absolutely no index images in the article. The authors do not cite them and completely in vain, without this it is difficult to assess the quality of subsequent processing of the results. The authors provided only one RGB image.
2. The authors assure that low-resolution images are not suitable, since both soil and other vegetation fall into the pixel. Then you need to somehow determine exactly how the values of the vegetation indices were averaged when shooting with high resolution, according to which criterion the pixel took or did not take part in the summation. Did the authors do it manually or was there some kind of algorithm? In Figure 7, the allocation of the main vegetation was carried out according to the NDVI index?
3. It is remarkable that the authors shot at the same time (9-00, 12-00), however, the indices are influenced not by the time of shooting, but by the angle of illumination from the Sun, which in July is not the same as in June.
4. The shooting mode is unclear - at the nadir, or at an angle.
Author Response
Dear reviewer
Thanks for the comments on the document.
The modifications have been highlighted in the manuscript.
I enclose the response to the comments
Kind regards
A wonderful article dedicated to the use of multispectral images to control plant parameters. However, a number of changes should be made to improve the quality of the presentation of the results.
There are absolutely no index images in the article. The authors do not cite them and completely in vain, without this, it is difficult to assess the quality of subsequent processing of the results. The authors provided only one RGB image.
Thank you for the comments. Based on your recommendations, we have added one more Figure showing the NDVI values for August 19th, of 2021 in the morning and noon (Figure 5 and lines 435 and 438). NDVI is chosen for visualization due to its widespread use and the well-known optimal values it can reach. We are not adding more figures for other indices as it would unnecessarily lengthen the document. However, If the reviewer feels that additional indices should be included, please do not hesitate to let us know again.
The authors assure that low-resolution images are not suitable, since both soil and other vegetation fall into the pixel. Then you need to somehow determine exactly how the values of the vegetation indices were averaged when shooting with high resolution, according to which criterion the pixel took or did not take part in the summation. Did the authors do it manually or was there some kind of algorithm? In Figure 7, the allocation of the main vegetation was carried out according to the NDVI index?
The methodology to select the pixels is described in lines 257 to 258: “A set of five points from the centremost area (top of the canopy) of each vine was selected to obtain each band reflectance value”
Regarding Figure 7 (we changed the number of this figure and now it is Figure 3), this figure is an explanation to get the percentage of covered soil (related to canopy development) using K-means algorithm in RGB images, any multispectral index has been used. This methodology is described in lines 223 to 228 “Canopy soil coverage was calculated from a high-resolution (3 cm/pixel) RGB image using QGIS software at the beginning of the irrigation season: 1 July 2021 and 30 June 2022. At this moment, the vegetative development stopped. The apex stops growing and does not develop more leaves. (QGIS, Free Software Foundation, Boston, MA, USA). A non-supervised k-means classification with two classes was run to differentiate vine canopy and soil pixels for a posterior percentage of covered soil calculation (Figure 3)”
It is remarkable that the authors shot at the same time (9-00, 12-00), however, the indices are influenced not by the time of shooting, but by the angle of illumination from the Sun, which in July is not the same as in June.
Thanks for the comment, we added this observation in lines 597-599: “However, another factor to consider in the evolution of the indices is the angle of illumination from the sun, it's not the same at the beginning of summer as at the end”
The shooting mode is unclear - at the nadir, or at an angle.
We added this information in line 250
Reviewer 3 Report
Comments and Suggestions for Authors
The research paper effectively discusses the application of imaging to assess vine water levels under different pruning methods. This new approach involves analyzing bands and Vegetation Indices (VIs) to link them with stem water potential and improve irrigation practices in vineyards. Key highlights include;
Utilizing UAV technology for large-scale assessments enables better water management and potentially boosts crop yields and quality through detailed spatial analysis of hydration levels within a vineyard.
Established connections between indices and plant hydration levels that could refine remote sensing techniques in precision viticulture.
Recommendations for research;
They are expanding the study to encompass a range of grape varieties and environmental factors to validate the spectral indices' effectiveness.
Exploring the incorporation of AI and machine learning methods to automate data analysis, further potentially enhancing accuracy and real-time application in settings.
They are increasing the frequency and diversity of data collection points to strengthen the model's capabilities and reliability across seasonal and climatic conditions.
Comments on the Quality of English LanguageMinor editing of English language required
Author Response
Dear reviewer
The authors express sincere gratitude for your comments and constructive feedback on our manuscript.
We have carefully considered each of your suggestions and have incorporated them into a chapter of future research in lines 684 to 689
Kind regards
Round 2
Reviewer 1 Report
Comments and Suggestions for Authors
The authors have considered all comments regarding the proposed corrections. The only thing that needs to be corrected in the manuscript is the following:
Line 399: repair the unit resolution (0.12 m/pixel) into 12 cm/pixel. Keep a uniform unit of 12 cm/pixel throughout the manuscript.
Author Response
Dear Reviewer
Thank you for your valuable feedback on our manuscript. We appreciate your attention to detail, as it has helped us identify areas for improvement.
we have made the necessary corrections about unit resolution in lines 399, 412, 600, 666, 706, and 715
We sincerely appreciate your time and effort in reviewing our manuscript.
Warm regards,
Reviewer 2 Report
Comments and Suggestions for Authors
The authors have made corrections to the text of the article based on all my comments. The article can be published in the presented form.
Author Response
Dear Reviewer
Thank you for your valuable feedback on our manuscript.
We sincerely appreciate your time and effort in reviewing our manuscript.
Warm regards